

# Classification of red beet and sugar beet for drought tolerance using morpho-physiological and stomatal traits

Gamze Kaya[1] and Nurgül Ergin[2]

[1] Department of Horticulture, Bilecik Şeyh Edebali University, Bilecik, Turkey
[2] Department of Field Crops, Bilecik Şeyh Edebali University, Bilecik, Turkey

## ABSTRACT

Drought is a global phenomenon that endangers agricultural production by creating water scarcity. Selecting drought-tolerant cultivars, varieties, and species is essential for maintaining the food supply and advancing breeding efforts. The study aimed to compare red beet (*Beta vulgaris* L. var. *cruenta*) and sugar beet (*B. vulgaris* L. var. *altissima* Döll.) for drought tolerance at the early growth stage considering morpho-physiological and stomatal parameters. Three red beet cultivars (Bicores, BT Pancina, and Yakut) and three sugar beet cultivars (Mohican, Orthega KWS, and Valentina) were subjected to various drought stress (Control, 10%, and 20% PEG-6000) for 30 days at the four-leaf stage. Fresh and dry plant weight, leaf area, dry matter, chlorophyll content (SPAD), leaf temperature, relative water content, membrane stability index, stomatal density, and size were investigated. The results revealed that the cultivars exhibited different responses to drought stress, and a greater percentage reduction in morphological parameters was observed in red beet cultivars. Drought markedly reduced the fresh and dry weights, leaf area, relative water content, membrane stability, and stomatal size. Enhanced dry matter and stomatal density were identified. The stomatal density increased from 158 to 215 $mm^{-2}$ while the stomatal size decreased from 433 to 342 $\mu m^2$ in the plants subjected to 20% PEG. Moderate drought stress effectively distinguished drought-tolerant sugar beet and red beet genotypes. It was concluded that sugar beet appeared to be more drought-tolerant than red beet and that the membrane stability index, relative water content, and stomatal density could be effectively used for selecting drought-tolerant beet genotypes.

# INTRODUCTION

The cultivated beets (*Beta vulgaris* L.) are categorized into four groups: leaf beet, garden beet (red beet), fodder beet, and sugar beet (*Lange, Brandenburg & De Bock, 1999*; *McGrath, Panella & Frese, 2011*). Among these groups, red beet (*Beta vulgaris* L. var. *cruenta*) is consumed as a vegetable because its roots are a source of minerals, antioxidants, sugars, dietary fibers, vitamins (A, C, D, E, and K), fatty acids, and natural pigments with antioxidant, anticancer, and radioprotective properties (*Escribano et al., 1998*; *Stagnari et al., 2014a*). It is increasingly popular for its use in cosmetics and natural dyes, and health

Corresponding author
Gamze Kaya,
gamze.kaya@bilecik.edu.tr

benefits because of its high content of potassium, magnesium, phosphorus, copper, and iron (*Akan, Tuna Gunes & Erkan, 2021*). Moreover, red beet helps prevent cancer, reduce blood pressure, support heart health, improve digestion, and aid in weight loss (*Suminarti, Dewi & Fajrin, 2020*; *Alavilli et al., 2023*). Sugar beet (*B. vulgaris* L. var. *altissima* Döll.) is another prominent beet variety, serving as the second source of refined sugar industry following sugarcane, accounting for approximately 30–40% of the world sugar supply (*Zhang, Nan & Yu, 2016*; *Hussein et al., 2019*; *Ghaffari et al., 2019*). Alongside its significance in the sugar industry, processed wastes and other byproducts are utilized in the manufacture of food additives, bioethanol, biodegradable polymers, and biofertilizers (*Magaña et al., 2011*; *Ghaffari et al., 2021*; *Yolcu et al., 2021*).

Drought, one of the primary constraints on global agricultural production, occurs when the soil water capacity falls to 12–20% for 16 days. This condition is influenced by several factors, including insufficient rainfall, salinity, extreme temperatures, high light intensity, and persistent water loss through transpiration or evaporation (*Hajheidari et al., 2005*; *Kränzlein et al., 2022*; *Bogati & Walczak, 2022*; *Alavilli et al., 2023*). Drought is a common problem in the arid and semiarid areas of Türkiye, where red beet and sugar beet are widely cultivated. Following sowing from mid-April to early May, insufficient rainfall leads to drought stress in young beet seedlings, adversely affecting their growth and development. The identification of tolerant cultivars, varieties, and species is critical for overcoming the adverse effects of drought and for recognizing genetic resources associated with drought tolerance for further breeding programs. Drought generally impacts various morphophysiological traits of beet varieties, including reduced germination performance (*Sadeghian & Yavari, 2004*), delayed and restricted fresh and dry weight (*Tan et al., 2023*), and reduced stomatal conductance (*Chołuj et al., 2014*), leaf area (*Suminarti, Dewi & Fajrin, 2020*), photosynthetic rate and transpiration (*Sattar et al., 2024*), and relative water content (*Romano et al., 2013*; *Zhang et al., 2024*), but increased dry matter (*Mahmoud et al., 2018*), leaf temperature (*Mohammadian et al., 2001*), relative chlorophyll content (SPAD) (*Zhou et al., 2023*; *Zhang et al., 2024*), and electrolyte leakage or membrane damage (*Sattar et al., 2024*; *Zhang et al., 2024*). Moreover, tolerance to drought is a complex quantitative characteristic influenced by multiple genes and associated with various physiological and biochemical processes that interact with environmental conditions (*Li et al., 2023*). For these reasons, it is important to identify drought-tolerant genotypes among the species that can be crossed with each other as donors in hybridization programs. This study aims to compare and classify the drought tolerance levels of red beet and sugar beet cultivars by examining their morphological, physiological, and stomatal characteristics.

## MATERIALS AND METHODS

This study was conducted in 2024 at the Department of Field Crops, Faculty of Agriculture, Eskişehir Osmangazi University, to compare sugar beet (*Beta vulgaris* L. var. *altissima* Döll.) and red beet (*Beta vulgaris* L. var. *cruenta*) under drought stress. Three sugar beet cultivars (Valentina, Orthega KWS and Mohican) and three red beet cultivars (Bicores, BT Pancina and Yakut) were used.

Seeds of sugar beet and red beet varieties were planted in seedling trays with a growing mixture of peat, perlite, and vermiculite (6:1:1, by volume) and placed in a growth chamber with a day/night temperature of 20 °C/10 °C, a photoperiod of 18 h during the day and 6 h at night, and a relative humidity of 65–70%.

## Drought conditions and plant growth

Two drought conditions were created by incorporating 10% (moderate) and 20% (severe) (w/w) polyethylene glycol (PEG)-6000 into the growing mixture, resulting in osmotic potentials of approximately 1.5 and 5.0 bar, respectively (*Michel & Kaufmann, 1973*; *Fan et al., 2022*). PEG-6000 was not added to the non-drought stress treatment as a control. When the seedlings reached the 4-leaf stage 21 days after sowing, they were transplanted into plastic pots filled with different PEG-6000-containing growing mixtures for subsequent plant growth. After transplanting, the plants were fertilized and irrigated with Hoagland's nutrient solution, and they were subsequently grown in a growth chamber at 22 °C day/18 °C night with an 18/6 h photoperiod. The pots were weighed every other day to detect water loss, and the soil moisture content was maintained by adding distilled water until the starting weight was reached. Harvest was performed thirty days after drought incubation. The percent reduction in morphological characteristics was determined using the formula described by *Rad & Abbasian (2011)*.

$$\% \text{ Reduction} = \left[ \frac{(\text{Value of control plant} - \text{Value of stressed plant})}{\text{Value of control plant}} \right] \times 100.$$

# MEASUREMENT OF MORPHOLOGICAL CHARACTERISTICS

The aerial parts of the plants were cut above the soil surface, and the fresh biomass was used to determine the fresh and dry weights after drying at 80 °C for 24 h. The leaves per plant were scanned and Image J software was used to calculate the leaf area (*Kaya, 2023*).

# MEASUREMENT OF PHYSIOLOGICAL CHARACTERISTICS

## Chlorophyll content

The chlorophyll content was measured at harvest with a portable chlorophyll meter, Konica Minolta SPAD-502 (Osaka, Japan), as the SPAD index. Three measurements were taken at different locations on the third leaf, and average values were calculated for each replicate.

## Leaf temperature

Leaf surface temperature was measured on the surface of the third leaf with infrared transducers (Trotec Model BP21).

## Relative water content

To assess the relative water content (RWC), two leaves (2nd and 3rd from the top) were collected from each replicate and weighed immediately to record the fresh weight (FW). The leaves were subsequently submerged in distilled water in a falcon tube for 24 h to reach turgidity. Following the quantification of turgor weight (TW), the leaves were subjected to

drying at a constant temperature of 80 °C for 24 h to ascertain the dry weight (DW). The RWC was calculated using the following formula described by *Kulan & Kaya (2024)*.

$$RWC\,(\%) = \left[ \left( \frac{FW - DW}{TW - DW} \right) \times 100 \right].$$

## Membrane stability index

The membrane stability index was employed to assess tissue damage resulting from drought stress. Ion leakage was measured on eight discs, each measuring four mm in diameter, excised from the third leaf. These discs were soaked in 30 mL of deionized water within a glass tube, which was allowed to incubate for 24 h at 20 °C in darkness. The first measurement of electrical conductivity ($EC_f$) was performed following the incubation period with an electrical conductivity meter (WTW 3.15i, Germany). The tubes were then floated in a water bath at 95 °C for 60 min to facilitate the release of electrolytes. The final measurement of electrical conductivity ($EC_l$) was performed after the tubes had cooled to room temperature. The membrane stability index (MSI) of the leaf tissue was determined according to *Aksu & Altay (2020)*.

$$MSI\,(\%) = \left[ \left( 1 - \frac{EC_f}{EC_l} \right) \times 100 \right].$$

## MEASUREMENT OF STOMATAL CHARACTERISTICS

The density and size of the stomata were assessed using the impression technique. The undersides of the third leaf were carefully coated with transparent nail polish at the center of the central vein and the leaf margin and then allowed to dry for 1–2 min. The dried varnish was carefully removed from the lamina. Stomatal density (stomatal number per $mm^2$ of leaf area) was quantified visually using a light microscope (Zeiss Axiophot microscope, $40 \times 10$) in conjunction with the image acquisition and digitizing program AxioVision 4.3 software, along with Canon EOS camera eyepieces. Three to five samples from each field were randomly picked from different areas of each sample and the counting process was repeated three times. The stomatal dimensions in the photographs were measured using an ocular micrometer calibrated with an object micrometer (*Kaya, 2021*).

The stomata size was computed by considering the width and length of stomata *via* the following formula (*Kaya, 2023*):

$$\text{Stomata size } (\mu m^2) = \left[ \frac{\text{Stomata width}}{2} \times \frac{\text{Stoma length}}{2} \right] \times \pi.$$

## Statistical analysis

The data were statistically analyzed for variance using the JMP 13.2 statistical program. Means were compared using Tukey's HSD test ($p < 0.05$). The experiment was established as a two-factor completely randomized design (CRD) with four replications.

**Table 1   Analysis of variance and main effects of cultivar and drought stress on fresh and dry weight, leaf area, dry matter, and chlorophyll content of red beet and sugar beet.**

| Factor | Fresh weight (g plant$^{-1}$) | Dry weight (g plant$^{-1}$) | Leaf area (cm$^2$) | Dry matter (%) | Chlorophyll content (SPAD) |
|---|---|---|---|---|---|
| **Drought (A)** | | | | | |
| Control | 12.60$^a$ | 1.12$^a$ | 250$^a$ | 9.9$^c$ | 43.2$^{a\dagger}$ |
| 10% PEG | 6.57$^b$ | 0.67$^b$ | 129$^b$ | 12.1$^b$ | 42.8$^a$ |
| 20% PEG | 3.54$^c$ | 0.37$^c$ | 55$^c$ | 12.9$^a$ | 41.5$^b$ |
| **Cultivar (B)** | | | | | |
| Bicores | 7.04b$^c$ | 0.63$^c$ | 132$^{bc}$ | 11.1$^c$ | 41.6$^{bc}$ |
| BT Pancina | 7.16$^b$ | 0.60$^c$ | 154$^a$ | 11.4$^{bc}$ | 39.1$^d$ |
| Yakut | 6.60$^c$ | 0.59$^c$ | 123$^c$ | 11.7$^{bc}$ | 41.2$^c$ |
| Mohican | 8.36$^a$ | 0.93$^a$ | 168$^a$ | 12.6$^a$ | 41.7$^{bc}$ |
| Orthega KWS | 8.03$^a$ | 0.84$^b$ | 139$^b$ | 11.9$^{ab}$ | 48.3$^a$ |
| Valentina | 8.24$^a$ | 0.78$^b$ | 154$^a$ | 11.0$^c$ | 43.2$^b$ |
| *Analysis of variance* | | | | | |
| A | ** | ** | ** | ** | ** |
| B | ** | ** | ** | ** | ** |
| A × B | ** | ** | ** | ** | ** |

Notes.

$\dagger$ Different superscript letters within each column refer to significance levels at $p < 0.05$.

** Significant at $p < 0.01$.

# RESULTS

An analysis of variance and the mean values of fresh and dry weight, leaf area, dry matter, and chlorophyll content of the sugar beet and red beet cultivars are presented in Table 1. The effects of beet cultivar and drought stress and their interactions were found to be significant. Drought stress greatly reduced the fresh and dry weights, leaf area, dry matter content, and chlorophyll content of the plants. Red beet cultivars presented lower fresh and dry weights than sugar beets did. Among the cultivars, Mohican exhibited the highest values of these characteristics, except for the chlorophyll content.

There were significant differences in leaf temperature, relative water content, membrane stability index, stomatal density, and size among drought stresses, cultivars, and their interactions (Table 2). Drought stress enhanced stomatal density and leaf temperature. Under increasing drought stress, enhanced stomatal density resulted in decreased stomatal size. Additionally, the relative water content and membrane stability index were lower in plants exposed to drought stress.

The changes in the fresh and dry weights of beet cultivars under drought stresses are shown in Fig. 1. Red beet and sugar beet cultivars presented similar fresh weights under control conditions, but heavier fresh weights were recorded for sugar beet cultivars under 10% PEG conditions (Fig. 1A). At 20% PEG, the differences observed between the cultivars disappeared. Fresh weight decreased by 66.6% in Yakut, 61.2% in BT Pancina, and 52.3% in Bicores at a 10% PEG concentration (Fig. 2). There was an apparent difference in dry weight between red beet and sugar beet cultivars. Under control and 10% PEG conditions, sugar

**Table 2 Analysis of variance and main effects of cultivar and drought stress on stomatal density, stomatal size, leaf temperature, relative water content, and membrane stability index of red beet and sugar beet.**

| Factor | Stomata density (number mm$^{-2}$) | Stomata size ($\mu$m$^2$) | Leaf temperature (°C) | Relative water content (%) | Membrane stability index (%) |
|---|---|---|---|---|---|
| **Drought (A)** | | | | | |
| Control | 158$^c$ | 433$^a$ | 22.5$^b$ | 83.2$^a$ | 83.8$^{a\dagger}$ |
| 10% PEG | 190$^b$ | 361$^b$ | 22.9$^a$ | 76.0$^b$ | 81.7$^b$ |
| 20% PEG | 215$^a$ | 342$^c$ | 23.3$^a$ | 65.8$^c$ | 71.5$^c$ |
| **Cultivar (B)** | | | | | |
| Bicores | 146$^d$ | 418$^a$ | 23.3$^{ab}$ | 74.1$^b$ | 73.9$^d$ |
| BT Pancina | 186$^c$ | 379$^{bc}$ | 22.6$^c$ | 79.1$^a$ | 81.8$^{ab}$ |
| Yakut | 182$^c$ | 371$^{bc}$ | 23.5$^a$ | 73.9$^b$ | 75.3$^d$ |
| Mohican | 197$^b$ | 381$^b$ | 22.4$^c$ | 79.2$^a$ | 83.8$^a$ |
| Orthega KWS | 218$^a$ | 368$^c$ | 22.7$^{bc}$ | 71.0$^c$ | 80.9$^b$ |
| Valentina | 197$^b$ | 355$^d$ | 22.9$^{abc}$ | 72.6$^{bc}$ | 78.3$^c$ |
| *Analysis of variance* | | | | | |
| A | ** | ** | ** | ** | ** |
| B | ** | ** | ** | ** | ** |
| A × B | ** | ** | ** | ** | ** |

Notes.
$^\dagger$ Different superscript letters within each column refer to significance levels at $p < 0.05$.
$^{**}$ Significant at $p < 0.01$.

beet cultivars exhibited increased dry weights, with Mohican demonstrating superiority over the other cultivars (Fig. 1B). However, these differences vanished at 20% PEG. With 10% PEG, the dry weight of the cultivar Yakut decreased by 57.3% (Fig. 2).

There were significant differences in leaf area between beet cultivars (Fig. 1C). Mohican had the greatest leaf area in both the control (285 cm$^2$) and 10% PEG (164 cm$^2$) groups. A greater percentage reduction in leaf area at 10% PEG was observed in the red beet cultivar Yakut (Fig. 2). However, Mohican was the cultivar most affected by drought stress at 20% PEG. The dry matter of the sugar beet cultivars was higher than that of the red beet cultivars (Fig. 1D). Drought caused a significant increase in the dry matter of red beet cultivars. However, no significant improvement in dry matter was observed between 10% and 20% PEG. In sugar beet cultivars, dry matter was enhanced at 20% PEG.

Drought stress significantly influenced the chlorophyll content of beet cultivars; however, no clear increase or decrease in chlorophyll content was detected (Fig. 3A). For sugar beet cultivars, the chlorophyll content increased at 10% PEG but decreased at drought stress of 20% PEG. In red beet cultivars, on the other hand, a drought level of 10% PEG reduced the chlorophyll content but slightly increased with 20% PEG. Furthermore, a meaningful increasing trend in leaf temperature was identified in Yakut and Valentina (Fig. 3B). The leaf temperature increased from 22.1 °C to 24.4 °C in Yakut and from 21.8 °C to 24.5 °C in Mohican, which may be a clue for the classification of cultivars on the bases of drought tolerance.

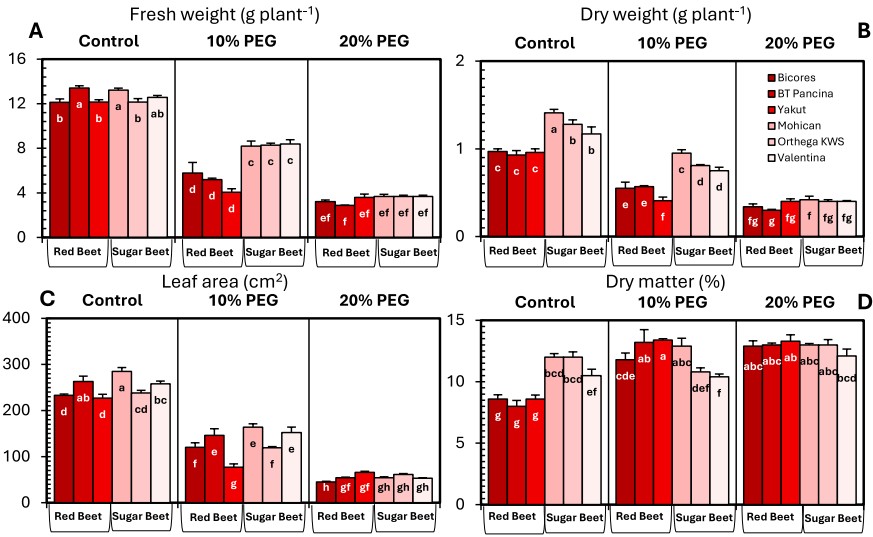

**Figure 1  Changes in fresh weight (A), dry weight (B), leaf area (C), and dry matter (D) of red beet and sugar beet cultivars exposed to increasing drought stress.** Bars on each column show standard error. Letters within each column denote significance levels at $p < 0.05$.

The relative water content of the sugar beet and red beet cultivars significantly decreased under drought stress (Fig. 3C). The sugar beet cultivar Orthega KWS presented the minimum relative water content (79.3%) under non-drought stress conditions. However, it resulted in the greatest percentage reduction at 10% PEG (Fig. 2). Compared to other cultivars, the red beet cultivar BT Pancina and the sugar beet cultivar Mohican presented greater relative water content under all levels of drought stress. Similarly, the membrane stability index was also higher in these cultivars. Drought stress decreased the membrane stability index, and the lowest value was recorded in Bicores and Yakut at 20% PEG (Fig. 3D). Under moderate drought conditions (10% PEG), the membrane stability index decreased by 12.2% in Valentina and 7.9% in BT Pancina (Fig. 2).

Drought stress resulted in an increase in stomatal density in both sugar beet and red beet cultivars (Fig. 4A). However, this trend was very apparent in Orthega KWS, whose stomatal density reached the maximum level under severe drought stress (Fig. 2). The highest stomatal density (260 number mm$^{-2}$) was recorded in the Orthega KWS at 20% PEG. Stomatal size differed with beet genotypes (Fig. 4B). Stomatal size declined under increasing drought stress, depending on the rise in stomatal density.

## DISCUSSION

There was significant variation in drought tolerance between sugar beet and red beet, even among cultivars, at the early growth stage. Increased drought stress led to reduced fresh and dry weights of sugar beet and red beet, while the cultivars showed different responses to drought stress. Compared with red beet cultivars, sugar beet cultivars produced greater fresh and dry weights under 10% PEG conditions; however, this superiority was neutralized at 20% PEG. This can be attributed to the inability of red beet and sugar beet to withstand

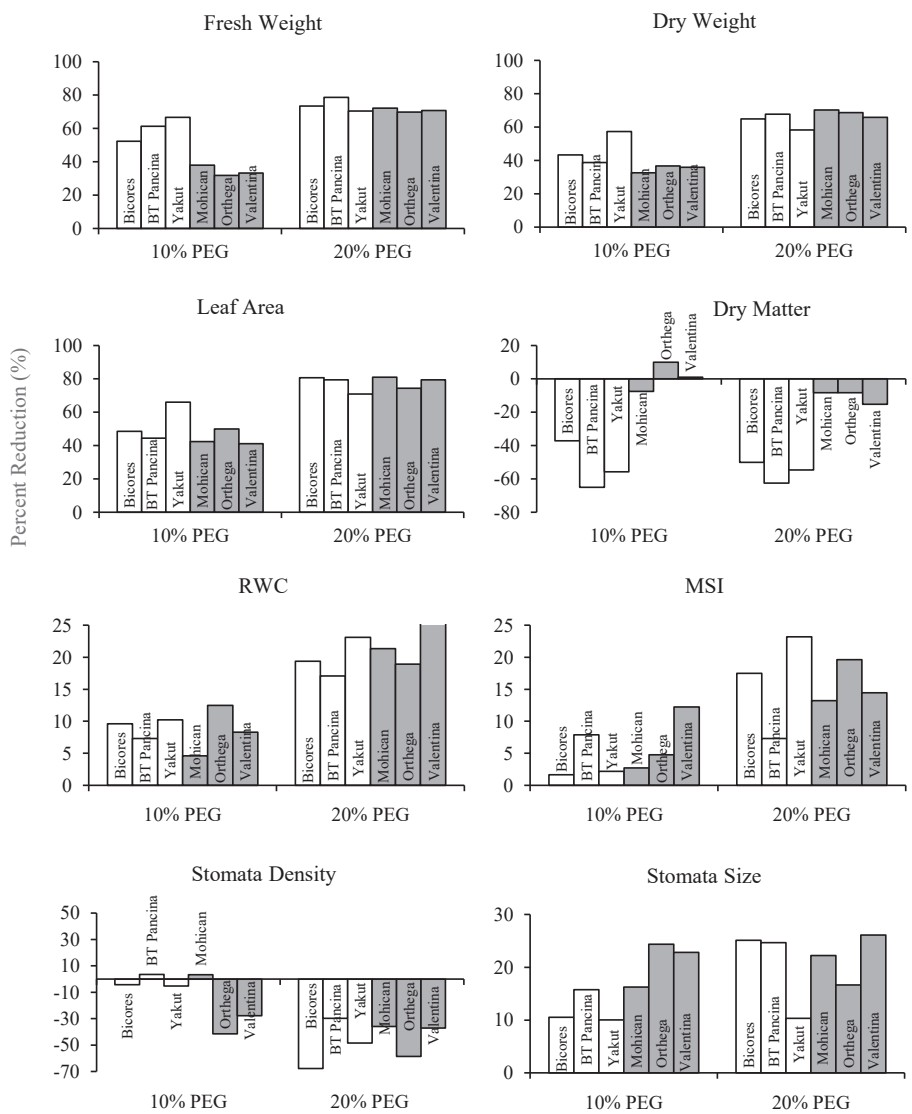

**Figure 2** The percentage reduction (%) in fresh weight, dry weight, leaf area, dry matter, relative water content (RWC), membrane stability index (MSI), stomata density, and stomata size of red beet and sugar beet cultivars subjected to drought stress induced by 10% PEG.

severe drought stress during the early growth stage. This result aligns with the findings of *Rad & Abbasian (2011)* in canola, who reported that significant differences between canola cultivars were observed under drought stress. Furthermore, the reduction in leaf area contributed to the decrease in the fresh and dry weights of red beet and sugar beet. Under moderate drought stress (10% PEG), the leaf area reduction reached the peak level of 66% in the red beet cultivar Yakut, followed by the sugar beet cultivar Orthega KWS with 50%. Our findings were corroborated by *Nelissen et al. (2018)*, who reported a 28% decrease in the leaf elongation rate of corn seedlings under mild drought conditions. *Stagnari et al. (2014b)* reported that the leaf area index (LAI) of red beet decreased by 39% with a water

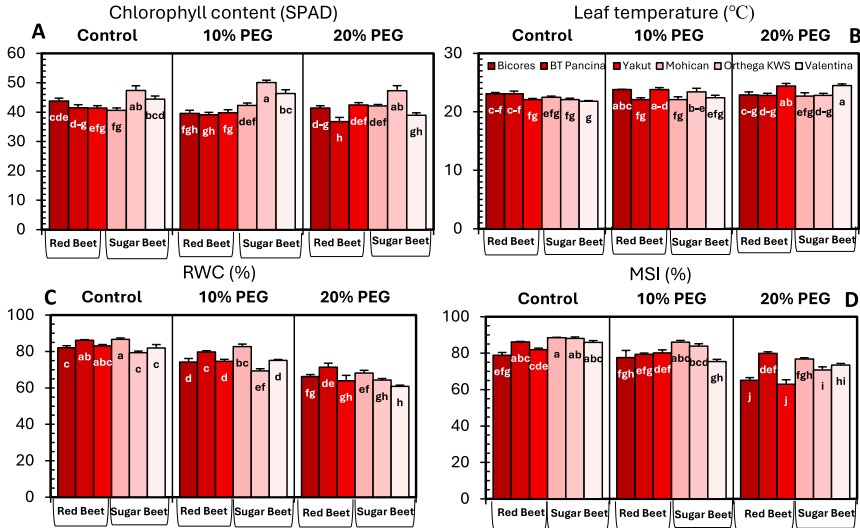

**Figure 3 Changes in chlorophyll content (A), leaf temperature (B), RWC (C), and MSI (D) of red beet and sugar beet cultivars exposed to increasing drought stress.** Bars on each column show standard error. Letters within each column denote significance levels at $p < 0.05$.

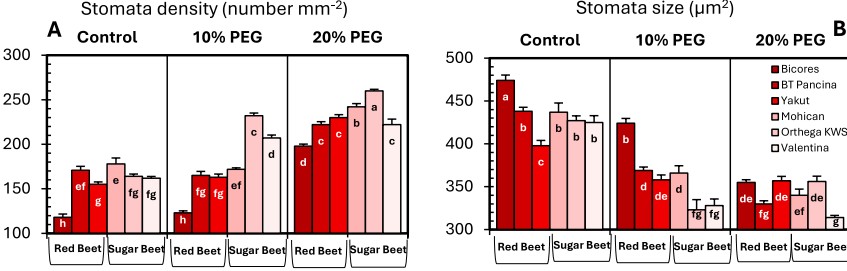

**Figure 4 Changes in stomatal density (A), and stomatal size (B) of red beet and sugar beet cultivars exposed to increasing drought stress.** Bars on each column show standard error. Letters within each column denote significance levels at $p < 0.05$.

supply of 50% and by 59% with a water supply of 30%. *Tan et al. (2023)* found that the fresh and dry leaf weights of sugar beet germplasm decreased with increasing levels (3%, 6%, and 9% PEG 6000) of drought stress in plants exposed for 2 and 7 days. *Suminarti, Dewi & Fajrin (2020)* indicated that the leaf area of sugar beet considerably decreased under water deficit conditions.

Dry matter accumulation is an important indicator of tolerance to several abiotic stresses. An apparent increase in dry matter in red beet cultivars under drought stress was observed, while a minimal increase was recorded in sugar beet cultivars. The increased dry matter accumulation in unstressed sugar beet plants may contribute to their drought tolerance. Our results confirm the findings of *Mahmoud et al. (2018)*, who reported that drought-tolerant genotypes produced higher dry matter than sensitive genotypes.

Drought stress strongly influences the chlorophyll content of beet cultivars, with increased drought stress leading to a decrease in the chlorophyll content of red beet cultivars. Among sugar beet cultivars, Valentina showed a decline in chlorophyll content, whereas Mohican presented an increase under drought conditions. *Basal, Szabó & Veres (2020)* reported that drought stress induced by PEG resulted in a lower relative chlorophyll content (SPAD) than the control in soybean. Similar results were reported in red beet by *Stagnari et al. (2014b)*, and in oat by *Xie et al. (2021)*.

Leaf temperature generally increases in plants subjected to various abiotic stresses and may be useful as a selection criterion because it increases in stressed plants (*Mohammadian et al., 2001*). In this study, the mean leaf temperature increased due to increased drought stress, but all the beet cultivars did not show a significantly similar trend. A linear increasing trend in leaf temperature was detected only in Yakut and Valentina, suggesting that these cultivars were more susceptible to drought stress than the other cultivars were. We argue that stomatal closure inhibits or ceases transpiration, raising the leaf temperature under drought stress as indicated by *Lourtie, Bonnet & Bosschaert (1995)* and *Stagnari et al. (2014a)*.

The relative water content of red beet and sugar beet cultivars decreased with increasing drought stress. A minimal reduction in relative water content was obtained in Orthega KWS and BT Pancina, which were the cultivars least affected by drought. The low relative water content of leaves is likely associated with high dry matter accumulation, resulting in improved drought tolerance. Additionally, decreased RWC and leaf area, along with increased dry matter, are correlated and contribute to drought tolerance in beet cultivars. This result is supported by *Chołuj et al. (2014)*, who suggested yield reduction in beets under drought resulting from changes in RWC and water potential in leaves, and *Bloch, Hoffmann & Märländer (2006)* determined restricted leaf expansion and $CO_2$ assimilation. Also, *Ober et al. (2005)*, *Aksu & Altay (2020)*, and *Zhang et al. (2024)* demonstrated that the relative water content increased in sugar beet plants exposed to drought stress.

Drought exposure leads to higher electrolyte leakage in the leaves, as it weakens the integrity of the cell membrane, making it less stable (*Wedeking et al., 2016*). Membrane stability serves as a valid criterion for assessing drought damage, with low ion leakage indicating high membrane stability. In the present study, significant differences in the membrane stability index were detected among the beet cultivars. Electrolyte leakage was greater in red beet cultivars than in sugar beet cultivars due to increasing drought stress; consequently, membrane stability decreased. BT Pancina and Mohican exhibited a relatively high membrane stability index at 20% PEG, suggesting that these plants maintain cell membrane stability under severe drought and show better tolerance to drought. This finding was supported by the results of *Rao et al. (2012)* and *Bijanzadeh, Barati & Egan (2022)*, who demonstrated that drought-tolerant maize genotypes presented low ion leakage, indicating high membrane stability. Similarly, *Sha et al. (2024)* and *Zhang et al. (2024)* reported increased electrolyte leakage in sugar beet plants exposed to drought stress.

Increased stomatal density and decreased stomatal size were observed in beet cultivars exposed to drought stress, but Orthega KWS presented the opposite response in that its stomatal density decreased with drought (Fig. 5). A similar trend was reported in barley

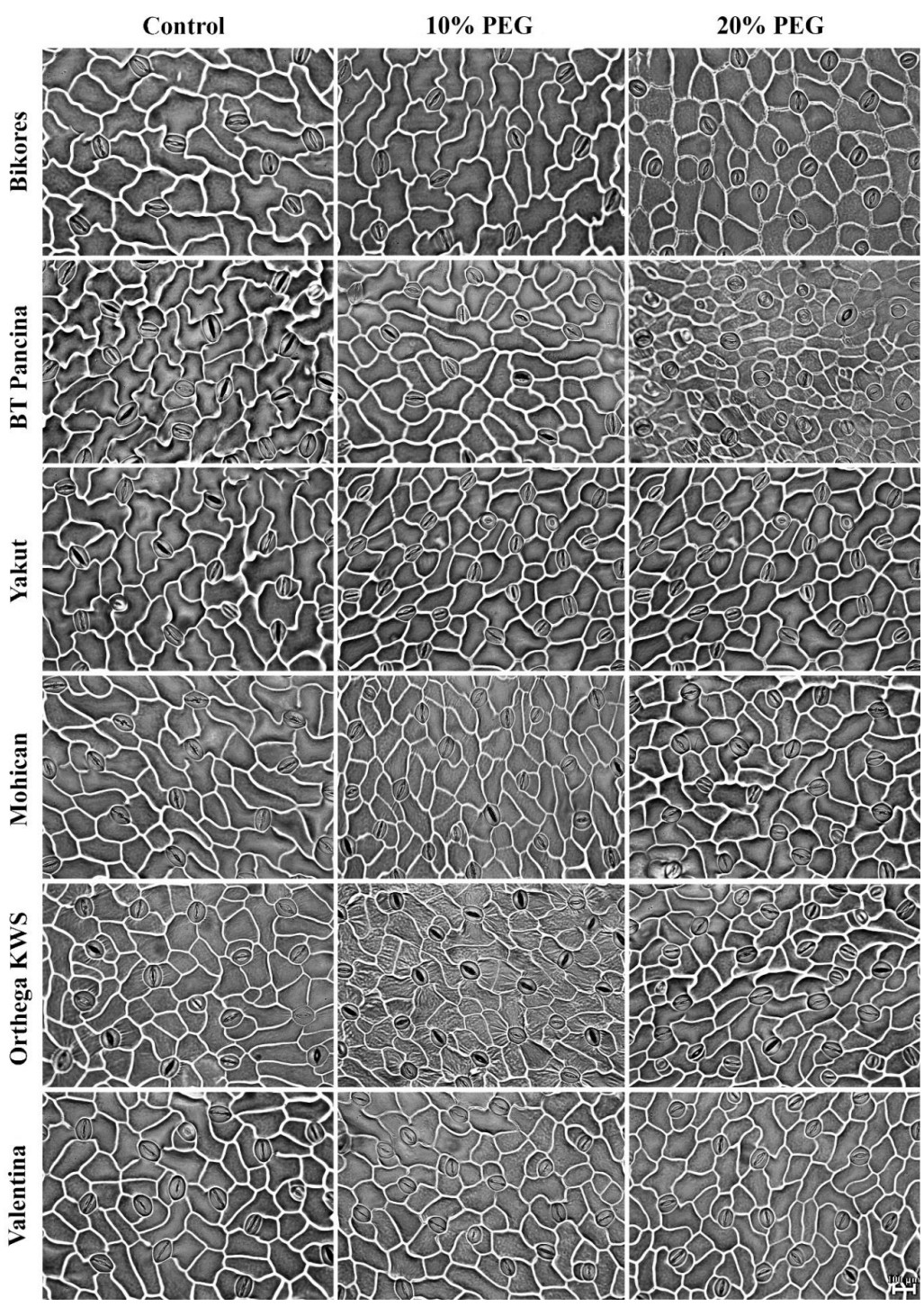

**Figure 5** **Stomata pictures of red beet and sugar beet cultivars plants under different drought stresses.**

by *Hughes et al. (2017)* and in safflower by *Ergin & Kaya (2023)* under salinity stress. Increased stomatal density and smaller stomatal size provide an adaptation to drought, as they enhance the ability of plants to regulate water transport and transpiration more efficiently. Low stomatal density in sugar beet is associated with water stress tolerance, as shown by *Luković et al. (2009)*, who reported that sugar beet varieties with stomatal densities between 70 and 150 number mm$^{-2}$ exhibited superior tolerance to water deficit compared to those with lower stomatal densities. Drought stress induces stomatal closure, reducing the size of stomatal pores to prevent water evaporation and maintain the water balance. These results indicate a decrease in stomatal size in drought-stressed plants, which is consistent with the findings of *Ferreira et al. (2024)*, who demonstrated that water deficit resulted in reduced cell expansion and stomatal closure in red beet, ultimately affecting beet root yield and quality. *Yolcu et al. (2021)* announced that stomatal closure and a reduction in RWC due to drought stress caused a decrease in photosynthetic activity and yield loss in sugar beets. Water use efficiency and water absorption from the different soil profiles with low water holding capacity vary with sugar beet genotypes and drought-tolerant ones could uptake much more water from deep soil layers (*Ober et al., 2005*). In addition, inhibiting water loss *via* transpiration by the closure of stomata contributes to drought tolerance. Stomatal closure or opening was not investigated in this study, while stomatal density and size were captured in Fig. 5.

## CONCLUSION

The classification of sugar beet and red beet cultivars according to drought tolerance at the four-leaf stage under controlled environmental conditions revealed that there was a clear difference between sugar beet and red beet. The variety or cultivar least affected by drought is considered tolerant. In this study, sugar beet cultivars appeared to be more tolerant to drought stress because of a low percentage reduction in morphological characteristics, membrane stability, and relative water content. Higher membrane stability was obtained from sugar beet cultivars, suggesting that the cell membrane was less injured by drought. Increased stomatal density was observed as a typical response to increasing drought stress and a lower percentage increase was identified in sugar beet cultivars compared to red beet cultivars. Moreover, there were significant differences in drought tolerance among the cultivars of sugar beet and red beet. The Orthega KWS presented the highest susceptibility to drought among sugar beet cultivars, whereas Yakut was among the red beet cultivars. Farmers should prefer drought-tolerant cultivars if they grow red beet or sugar beet, with a preference for sugar beet in regions where drought is a common problem. In conclusion, red beet cultivar BT Pancina and sugar beet cultivar Mohican should be recommended for cultivation under drought conditions and as genetic resources for breeding research. In addition, further studies are needed to confirm these results and to classify beet cultivars for drought tolerance under field conditions.

### Funding

The authors received no funding for this work.

### Competing Interests

The authors declare there are no competing interests.

### Author Contributions

- Gamze Kaya conceived and designed the experiments, performed the experiments, analyzed the data, prepared figures and/or tables, authored or reviewed drafts of the article, and approved the final draft.
- Nurgül Ergin performed the experiments, analyzed the data, prepared figures and/or tables, authored or reviewed drafts of the article, and approved the final draft.

### Data Availability

The raw data are available in the Supplemental File.

### Supplemental Information

Supplemental information for this article can be found online at http://dx.doi.org/10.7717/peerj.19133#supplemental-information.

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
