# Peer review of "Classification of red beet and sugar beet for drought tolerance using morpho-physiological and stomatal traits"

_PeerJ, doi:10.7717/peerj.19133_

## Round 0.1 · original submission · Minor Revisions

Dear authors

Your manuscript needs minor revisions before the acceptance. The reviewers addressed all points. Your manuscript is scientific and statistically sound.

Reviewer 1 ·

Basic reporting

* Line 49 is written in somewhat informal. It should be in scientific way.
* Line 107: Proper units is to mentioned.
* English language used is to be checked as in professional way.

Experimental design

* Table 1 and Table 2 are not clearly explained. Please make it understandable in comparison's.
* Research question are well defined and relevant.

Validity of the findings

* Data provided are robust and statistically sound.
* Conclusion part should be linked more to the supporting results. Add some more concluding supports.

·

Basic reporting

1.1 Language: The manuscript demonstrates professional use of language, but certain sections require refinement for clarity, particularly in the introduction and results sections. Technical jargon can be simplified for readers unfamiliar with plant physiology.

1.2 Introduction: The introduction provides a strong background on the study’s relevance. However, it should clearly specify the knowledge gap being addressed by the research. Expanding on the study’s objective would improve the narrative.

1.3 References: The references cited are current and relevant, covering both foundational and recent studies. The authors have demonstrated a solid understanding of prior research.

1.4 Figures and Tables: Figures and tables are relevant and well-organized, but some captions lack sufficient detail. For example, Figure 2 should include more explanation of what the stomatal images signify in terms of drought tolerance.

Experimental design

2.1 Research Question: The study’s research question is clearly defined, focusing on the comparative drought tolerance of different beet cultivars. It aligns well with the scope of the journal.

2.2 Methodology: The methods are detailed and reproducible. However, the rationale for selecting the 10% and 20% PEG-6000 drought stress levels needs further elaboration. Justifying these levels with prior studies or preliminary data would strengthen this section.

2.3 Parameters: The selected parameters—fresh and dry weight, relative water content, stomatal density, and membrane stability—are appropriate and scientifically valid for assessing drought tolerance.

2.4 Ethical Standards: The research adheres to ethical and technical standards expected in plant studies.

Validity of the findings

3.1 Data Presentation: The data are presented comprehensively, with appropriate statistical analyses. However, there is redundancy between the narrative text and data tables. Condensing repetitive information would improve readability.

3.2 Statistical Analysis: The use of analysis of variance (ANOVA) and Tukey’s HSD test is appropriate for comparing the effects of drought on various traits. Statistical significance is clearly indicated.

3.3 Interpretation: The discussion provides a solid interpretation of the results but could further explore potential genetic or physiological mechanisms explaining the superior drought tolerance of sugar beet.

3.4 Relevance: The findings are relevant to both researchers and agricultural practitioners. Highlighting practical implications, such as how these results can inform breeding programs, would increase the study’s impact.

Additional comments

4.1 Strengths: The manuscript’s strengths include its comprehensive dataset, examination of multiple drought-related traits, and comparison of several cultivars. The findings are timely and relevant given the global challenge of drought stress in agriculture.

4.2 Improvements:
- Refine the language to enhance clarity and accessibility for a wider audience.
- Provide more detailed figure captions to improve the reader’s understanding of the visual data.
- Reduce redundancy in the presentation of data across tables and text.
- Expand the discussion on the genetic and physiological traits underlying drought tolerance.

4.3 Practical Implications: The conclusion should include actionable recommendations for farmers and breeders. Discussing the broader applicability of the findings would enhance the article’s practical relevance.

Reviewer 3 ·

Basic reporting

The manuscript is well written in English, but some minor errors should be corrected. For instance, in the Abstract section, in Line 24, … sugar beet (B. vulgaris L. var. altissima Döll.) and for drought tolerance…” “and” should be deleted. Please reconsider it.

The introduction section provides sufficient information on drought stress and general plant responses. In addition, the effects of drought stress on sugar beet plants are well documented in the introduction, but the novelty of the research is that no specific knowledge is provided about the response of red beet to drought and its comparison with sugar beet.

The figures and tables are relevant and well-designed. However, tables showing the interaction effects of drought stress and beet varieties should be converted to figures. Standard errors should be added to mean values, and the captions should be more self-explanatory.

Experimental design

The research question is well-defined and concentrates on the comparative drought tolerance of different beet cultivars.

The methodology is valid for the aims of the research.

Validity of the findings

The data are presented in the Tables with appropriate statistical comparisons. However, the interactive effects may be presented in figures.
To determine the effects of drought on the target parameters, analysis of variance (ANOVA) and Tukey’s HSD test are appropriate. Standard errors should be connected to mean values.

The results are important for farmers and breeders to improve new drought varieties. The study identified valuable selection criteria for drought in red beet and sugar beet, and the parameters such as fresh and dry weight, relative water content, stomatal density, and membrane stability are suitable for evaluating drought tolerance.

Additional comments

The manuscript classifies red beet and sugar beet cultivars using various morphological, physiological, and stomatal traits according to their drought tolerance. Drought-related traits are identified, and suggestions for farmers and breeders are made. The findings are relevant and novel.
Minor language editing should be recommended.
Figure and table captions should be detailed.

---

## Round 0.2 · accepted · Accept

Your manuscript accepted. Congralulations